# Mitochondrial Dynamics Drive Muscle Stem Cell Progression from Quiescence to Myogenic Differentiation

**DOI:** 10.3390/cells13211773

**Published:** 2024-10-26

**Authors:** Olivia Sommers, Rholls A. Tomsine, Mireille Khacho

**Affiliations:** 1Department of Biochemistry, Microbiology and Immunology, Faculty of Medicine, University of Ottawa, Ottawa, ON K1H 8M5, Canada; 2Center for Neuromuscular Disease (CNMD), University of Ottawa, Ottawa, ON K1H 8M5, Canada; 3Ottawa Institute of Systems Biology (OISB), Faculty of Medicine, University of Ottawa, Ottawa, ON K1H 8M5, Canada

**Keywords:** mitochondria, mitochondrial dynamics, muscle stem cells, OPA1, DRP1, reactive oxygen species (ROS), glutathione, differentiation, metabolism, myogenesis

## Abstract

From quiescence to activation and myogenic differentiation, muscle stem cells (MuSCs) experience drastic alterations in their signaling activity and metabolism. Through balanced cycles of fission and fusion, mitochondria alter their morphology and metabolism, allowing them to affect their decisive role in modulating MuSC activity and fate decisions. This tightly regulated process contributes to MuSC regulation by mediating changes in redox signaling pathways, cell cycle progression, and cell fate decisions. In this review, we discuss the role of mitochondrial dynamics as an integral modulator of MuSC activity, fate, and maintenance. Understanding the influence of mitochondrial dynamics in MuSCs in health and disease will further the development of therapeutics that support MuSC integrity and thus may aid in restoring the regenerative capacity of skeletal muscle.

## 1. Introduction

### 1.1. Muscle Stem Cells

The plasticity of skeletal muscle can be attributed to the function of adult muscle stem cells (MuSCs, also known as satellite cells). This resident stem cell population, which is found beneath the basal lamina of mature myofibers, is responsible for muscle regeneration and the maintenance of muscle mass and quality [1,2]. Under homeostatic conditions, MuSCs exist in a quiescent state [3]. Quiescence is a state of reversible cell cycle arrest, and in MuSCs, it is marked by the expression of the transcription factor Pax7, along with a number of cell surface markers, such as CD34 [4,5,6,7]. As stem cells, MuSCs are defined by their ability to commit to differentiation or to self-renew and replenish the MuSC pool [8,9]. Upon cell-intrinsic or -extrinsic stimuli that trigger the initiation of myogenesis, such as those associated with muscle injury, MuSCs exit quiescence, activate, undergo commitment to a myogenic fate, and differentiate into multinucleated myotubes. Myogenic commitment and differentiation are mediated through the progressive expression of the myogenic regulatory factors (MRFs), referred to as myogenic factor 5 (Myf5), myogenic differentiation 1 (MyoD), and myogenin (MyoG) [10,11], coinciding with the downregulation of Pax7 and transcription factors SNAI1/2, which were found to regulate proliferating myoblasts’ entry into differentiation [11,12,13,14,15,16]. Myoblast differentiation into myocytes and their fusion into myotubes is regulated by several MRFs, including MyoG and MRF-4 (which promote both commitment and differentiation) [17,18,19,20].During differentiation, myogenic progenitors upregulate MyoG and MRF-4 [21,22] and undergo fusion to form new myofibers which transiently express myosin heavy chain (eMHC) [23,24,25]. As the fibers regenerate and fully develop, adult isoforms of MHC are expressed [26,27,28]. Conversely, self-renewing MuSCs, represented by the retention of Pax7 expression, return to a state of quiescence [29]. The balance between commitment and self-renewal is essential for efficient muscle regeneration and ensuring the maintenance of a stem cell pool that is available to maintain muscle homeostasis.

A diverse range of factors are responsible for regulating myogenesis; however, both metabolic and mitochondrial functions have recently been shown to drive MuSC quiescence, activation, and commitment [25,30,31,32,33,34,35,36]. Mitochondrial dynamics have emerged as a point of interest in the field of regenerative medicine, owing to their significant role in modulating stem cell activity, fate, metabolism, and cell signaling, among other diverse functions. In this review, we discuss the relevance of mitochondrial dynamics as a regulator of MuSC activity and fate decisions.

### 1.2. Mitochondrial Dynamics

Mitochondria are highly dynamic, double-membrane organelles that have critical and diverse roles within the cell. Through mitochondrial dynamics, mitochondria constantly modify their morphology, allowing them to maintain proper function and respond to stimuli [37,38]. A balance between mitochondrial fission and fusion is achieved in an ever-shifting, tightly regulated system. In this way, mitochondrial dynamics play an integral role in maintaining the integrity of the mitochondrial network. Mitochondrial dynamics are driven by a dynamin-related family of large GTPases [39]. Using the energy from GTP hydrolysis, these enzymes perform mechanical work on the two membranes of mitochondria to promote their fragmentation or fusion [37]. While mitochondrial fusion facilitates the distribution of metabolites and mitochondrial proteins or the functional complementation of dysfunctional components throughout the network, fission can segregate those components and also target damaged mitochondria for removal via mitophagy [40]. Both fission and fusion are essential to mitochondrial homeostasis, and the balance between them is key.

#### 1.2.1. An Overview of Mitochondrial Fission

The initiation of fission involves the polymerization of actin and the contact of endoplasmic reticulum (ER) tubules with mitochondria to constrict the outer mitochondrial membrane (OMM) [41]. The fission GTPase, DRP1 (Dynamin-related protein 1), exists in the cytoplasm in its inactive form, and its activation is mediated by post-translational modifications, such as phosphorylation, SUMOylation, or ubiquitination [42,43]. Once active, DRP1 is translocated from the cytoplasm to the OMM and physically interacts with adaptor proteins [42]. In mammals, there are four DRP1 adaptors, including Fis1 (Mitochondrial fission 1), MFF (Mitochondrial fission factor), and MID49/MID51 (Mitochondrial dynamics protein 49 and 51) [37,44]. Upon the oligomerization of DRP1 into a ring-like complex at the fission site on the mitochondrial surface, the membrane constricts, and the mitochondrion is fragmented into two distinct organelles [39,45] (Figure 1).

#### 1.2.2. An Overview of Mitochondrial Fusion

Mitochondrial fusion occurs sequentially on both membranes, beginning with the fusion of the two OMMs and then the inner mitochondrial membranes (IMMs). MFN1 (Mitofusin 1) is the dynamin-related GTPase that mediates OMM fusion [46]. MFN2 (Mitofusin 2) is the homolog of MFN1 that can mediate mitochondrial fusion but has distinct roles in processes such as ER–mitochondria tethering [47,48,49]. The GTP hydrolysis of MFNs induces a conformational change that allows two distinct OMMs to contact one another and fuse together [50]. OPA1 (Optic atrophy 1) modulates fusion at the IMM and also influences the cristae architecture and integrity within the mitochondria through its remodeling functions [51,52,53] (Figure 1).

The modulators of mitochondrial dynamics described above are essential for proper development and function. For instance, complete knockouts of DRP1, OPA1, or MFN1/2 have been shown to be lethal during embryonic development in mice by inducing mitochondrial dysfunction [46,54,55,56]. Additionally, knockouts of the fusion modulators MFN1/2 and OPA1 lead to reduced respiration rates, altered cellular metabolism, loss of membrane potential, and reduced cellular growth rates [53,57,58,59]. Thus, mitochondrial dynamics play an essential role in cellular health and function. In the sections below, we describe the ways in which mitochondrial dynamics influence MuSC activity, fate, and function throughout myogenesis.

## 2. Mitochondrial Dynamics in Mediating MuSC Quiescence and Activation

The morphology of mitochondria is distinct between the cell populations spanning the various phases of myogenesis [25,60] (Figure 2). In quiescent MuSCs, mitochondria exist in a relatively elongated morphology, which becomes fragmented upon MuSC activation [25]. The length of mitochondria influences the depth of MuSC quiescence, with elongated mitochondria being associated with a deeper quiescent state and more fragmented mitochondria priming MuSCs for activation in a state known as G_alert_. During activation, the expression of OPA1 declines [25], and DRP1 increases [33], enabling a more fragmented morphology. In particular, it has been shown that HGF/mTOR signaling is required for physiological fragmentation of mitochondria during MuSC activation [25]. Interestingly, activation of the mTOR pathway, as seen by pS6 (phosphorylated 40S ribosomal protein S6) expression, is an established marker of the G_alert_ state [61]. HGF-mediated activation of mTOR was shown to promote MuSC activation through mitochondrial fission [25]. Conversely, by inhibiting mTOR with rapamycin, mitochondrial fragmentation is prevented, and the quiescent state of MuSCs is maintained [25]. Models of increased mitochondrial fission, such as the MuSC-specific knockout of OPA1 (OPA1-KO), exhibit a significant increase in the mTOR pathway constituent, pS6, showing that these MuSCs have become primed for activation in a G_alert_ state and that mitochondrial dynamics serve as a mediator for MuSC activation [25]. This research highlights the physiological need for temporally regulated and transient fragmentation of mitochondria to guide MuSCs out of quiescence and into a G_alert_ or fully activated state.

Beyond precocious activation, loss of OPA1 is associated with enhanced kinetics of cell cycle entry and commitment in MuSCs [25]. In comparison, MuSCs from a transgenic OPA1 overexpression model (OPA1-OE) exhibit more elongated mitochondria and delayed kinetics of activation [25]. In related work, the role of mitochondrial fission in MuSCs was explored in a MuSC-specific knockout of DRP1 (DRP1-KO) to find that mitochondrial fragmentation is essential for the MuSC proliferation that enables muscle regeneration [33]. The transcriptome of freshly isolated DRP1-KO MuSCs, with hyper-elongated mitochondria, was shown to mimic that of activated, rather than dormant, MuSCs and was equally enriched for electron transport chain (ETC) and tricarboxylic acid (TCA) cycle genes [33]. As such, DRP1-KO MuSCs are transcriptionally primed for activation but do not precociously exit quiescence in resting muscle [33].

The mitochondrial fragmentation associated with MuSC activation coincides with a physiological increase in mitochondrial reactive oxygen species (mtROS). In MuSCs, glutathione (GSH) contributes in coordination with mtROS to influence activation, fate decisions, and self-renewal [25]. Through the acute loss of OPA1 in MuSCs, the abundance of GSH and the redox marker SLC7a11, a cysteine/glutamate antiporter, as well as expression of the GSH biosynthesis genes *GSS* and *GLCL* increases, alongside the physiological increase in mtROS during activation [25]. In a similar manner, treating muscle cells with GSH or low doses of rotenone to stimulate Complex-I-mediated mtROS generation is sufficient to downregulate transcript levels of *Pax7* and *CD34* while equally upregulating *MyoD* and *MyoG* gene expression, indicative of an exit from quiescence and a commitment to differentiation [25]. Comparatively, *MyoD* and *MyoG* expression was lower in cells treated with low doses of the DRP1 inhibitor Mdivi-1, the mitochondrial antioxidant mitoTEMPO, or BSO that depletes GSH levels [25]. Of note, it is important to consider that Mdivi-1 at high doses is a Complex-I inhibitor [62]. Therefore, only low doses can be utilized for proper evaluation of the effects of mitochondrial fission inhibition, which Baker et al. respected [25,62]. This finding highlights a new role for GSH as a cell signaling mediator in influencing the transcription of myogenic regulatory factors and raises questions regarding the diverse ways in which the redox status of the cell can influence cell signaling programs.

Together, the studies discussed above indicate that by modifying mitochondrial morphology, the depth of MuSC quiescence can be regulated. Mitochondria are also key for the ability of MuSCs to respond to environmental cues, such as through redox signaling. It would be of great interest to further understand the mechanisms underlying OPA1 and DRP1 regulation, among that of other dynamics regulators, during the transition from quiescence to activation to inform the development of therapeutics for neuromuscular disorders associated with stem cell depletion or exhaustion.

## 3. Mitochondrial Dynamics in Mediating MuSC Cell Fate and Differentiation

Just as mitochondrial dynamics are critical for the exit of quiescence and activation, dynamics are crucial for MuSC fate decisions. The balance between self-renewing and committing MuSCs in this process is essential to ensuring skeletal muscle health and function [63]. Mitochondrial dynamics and myogenesis are intimately linked, although not all the roles of mitochondrial shaping machinery are known in the context of MuSC commitment, self-renewal, and differentiation. OPA1 and DRP1 have been the most extensively studied, and below, we present findings that underscore their utmost importance during these cellular processes.

### 3.1. Commitment and Self-Renewal

Previous literature has shown that mitochondrial fragmentation promotes stem cell commitment [64,65,66,67,68]. Interestingly, proliferating MuSCs are represented as heterogenous populations of committing (Pax7-low) and self-renewing (Pax7-high) cells [29,69]. Mitochondrial morphological studies of cells destined for self-renewal reveal that mitochondria would resume their elongated morphology and reestablish the mitochondrial network pattern [25]. In the recent literature, it has been shown that the genetic knockout and pharmacological inhibition of OPA1 lead to mitochondrial hyperfragmentation-induced dysregulation in the balance of MuSC fate decisions [25]. Specifically, Baker et al. found that a significant majority of OPA1-KO MuSCs committed to the myogenic lineage at the expense of self-renewal [25]. Additionally, the pharmacological inhibition of mitochondrial fission with Mdivi-1 treatment in OPA1-KO cells was sufficient in rescuing the enhanced commitment phenotype and reestablished the balance between commitment and self-renewal [25]. Furthermore, mitochondrial dynamic-dependent changes in glutathione/redox states were revealed to mediate myogenic commitment [25]. To show this, Baker et al. treated primary MuSCs with compounds that alter mtROS and GSH levels. Interestingly, they observed enhanced myogenic commitment when MuSCs were treated with agents that moderately increase mtROS, such as low doses of rotenone or mito-Paraquat. Treatments with exogenous GSH or the GSH precursor cysteine yielded similar results. All cases mimicked the OPA1-KO MuSC phenotype of enhanced commitment, suggesting that these redox signaling mechanisms function downstream of the mitochondrial shaping protein. To further corroborate their findings, the opposite phenotype was observed when mtROS levels were dampened, and GSH levels were depleted. Indeed, under such circumstances, MuSCs tended to undergo more self-renewal, as seen through an increased number of Pax7^+^ nuclei. These findings highlight the importance of not only OPA1 but also the intimate role between mitochondrial dynamics and MuSC commitment and fate.

### 3.2. Differentiation

Differentiation is crucial for proper skeletal muscle tissue regeneration. As shown in the earlier stages of myogenesis, mitochondrial dynamics indeed play a crucial role, and this trend continues with differentiation. Recently, it was observed that both mitochondrial network and cristae remodeling occur during early and late stages of differentiation [25]. Specifically, the mitochondrial network becomes more extensive and prominent during the process of differentiation (Figure 3) [25,60]. Here again, mitochondrial dynamics machinery, including DRP1 and OPA1, have been shown to be key players in promoting myogenic differentiation. For example, De Palma et al. proposed an intriguing possible mechanism which states that mitochondrial elongation through the endogenous NO/cGMP-dependent (nitric oxide/cyclic guanosine monophosphate) inhibition of DRP1 during the process of myogenic differentiation is a necessary step for proper differentiation [70]. Interestingly, myogenic differentiation triggered the generation of NOS-dependent cGMP through the activation of guanylate cyclase. NO/cGMP signaling activated G-kinase, which inhibited DRP1 activity and translocation by phosphorylating its Ser637 residue [70]. The inhibition of DRP1 through NO/cGMP signaling shifted mitochondrial dynamics in favor of elongation, and during early differentiation, the mitochondrial morphology shifted from round and fragmented to elongated and extensive networks [70]. Additionally, NO is a known reversible inhibitor of cytochrome c oxidase, allowing it to modulate mitochondrial respiration [70,71,72]. Impairments in NO/cGMP signaling were associated with changes in mitochondrial membrane potential and increased sensitivity to oxidative stress, ultimately contributing to mitochondrial dysfunction [70]. This suggests that during differentiation, the NO/cGMP-mediated transient inhibition of DRP1, and, by consequence, mitochondrial elongation, may serve a protective role against oxidative stress. To further corroborate these results, Triolo et al. elegantly confirmed that OPA1 was essential for muscle regeneration, proper mitochondrial function, and differentiation in both C2C12 myoblasts and primary murine myoblasts. Furthermore, they uncovered that mitochondrial morphology, cristae architecture, and ATP levels were altered in a biphasic manner, where the initial changes occurred as early as 6 h post-differentiation, followed by additional remodeling and a rise in ATP levels at 48 h and 72 h. Specifically, they observed increased cristae biogenesis and cristae tightening, which is commonly associated with enhanced mitochondrial function [57,73,74,75,76]. There was evidence of a time-dependent increase in the oxygen consumption rate (OCR) and the upregulation of electron transport chain (ETC) proteins, Complex-I (NDUFA9), Complex-II (SDHA), Complex-III (UQCRC2), Complex-IV (MTCO1), and Complex-V (ATP5a) during differentiation. This led to the investigation of the master regulator of mitochondrial supercomplex formation and respiratory efficiency, SCAF1 (supercomplex assembly factor 1; also known as Cox7a21) [77,78,79]. In the absence of OPA1 and SCAF1, myoblasts could not undergo critical metabolic reprogramming nor differentiation [60]. This serves to further underscore the importance of mitochondrial bioenergetic and ultrastructural changes during the first days of differentiation. Furthermore, work carried out by Sin et al. found that a transient increase in DRP1 was required for proper differentiation in C2C12 myoblasts [80]. Specifically, an increase in DRP1 rapidly followed by mitophagy prior to the increased expression of OPA1 led to the reformation of the mitochondrial network and proper differentiation [80]. In line with this thinking, Kim et al. showed that DRP1 expression is upregulated and that DRP1 is rapidly translocated to the mitochondria within the initial days of myogenic differentiation [81]. Bloemberg et al. further corroborated these findings, showing that the inhibition of DRP1 using Mdivi-1 leads to a notable reduction in MyoG and myosin expression and abolishes the formation of myotubes [82]. However, these results are subject to interpretation as high doses of Mdivi-1 were used. Thus, it is the continuous shifts in expression levels between fission and fusion machinery leading to the constant restructuring of the mitochondrial network, along with remodeling cristae architecture, that allows for differentiation.

It has become very well understood that for myogenic differentiation to occur, cells must undergo metabolic reprogramming and switch their predominant method of ATP production from glycolysis to mitochondrial oxidative phosphorylation (OXPHOS) [25,30,83,84,85,86,87,88,89,90]. Recently, it has been found that OPA1 indeed is essential for the metabolic switch that is necessary for myogenic differentiation [60]. Genetic knockout or siRNA knockdown of OPA1 and the MYLS22-mediated pharmacological inhibition of OPA1 were implemented to find that OPA1 induces a metabolic switch during differentiation via the regulation of SCAF1 [60]. These findings underscore the importance of balanced cycles of fission and fusion and further elucidated the mechanisms involved for future therapeutic targeting, although interventions that target mitochondrial dynamics would have to be finely tuned.

The models above show that a complete loss of the modulators of mitochondrial dynamics, OPA1 and DRP1, impair differentiation (Table 1). However, in a physiological setting, it is the temporal and transient regulation of these factors that allows mitochondria to respond through changes in their morphology and cellular metabolism. Thus, there is a need for models that more accurately represent this finely tuned downregulation rather than a complete loss of expression or function. For instance, patient-derived loss of function mutations present an excellent opportunity to explore the nuances involved in the regulation of mitochondrial dynamics [91].

## 4. Perspectives

### 4.1. Role of MFN1/2

Currently, the roles of DRP1 and OPA1 in MuSCs, although not completely understood, are continuing to be discovered. However, other crucial mitochondrial dynamics machineries, such as MFN1 and MFN2, remain to be fully understood in the context of MuSCs [46]. Although they have some structural similarities due to their dual homotypic and heterotypic nature, these two enzymes can operate individually and have been shown to have distinct roles during various physiological and pathological conditions [46,48]. For example, MFN2 overexpression was found to not only be non-pathogenic but also to promote muscle hypertrophy in both young and old mice [92]. Currently, in skeletal muscle, one of the roles of MFN2 involves mediating sarcoplasmic reticulum–mitochondria crosstalk and calcium (Ca^2+^) uptake [93]. In skeletal muscle, the optimal mitochondrial Ca^2+^ uptake and buffering of myoplasmic Ca^2+^ requires proper mitochondrial localization, membrane potential, and morphology, which MFN2 was found to mediate [93]. In C2C12 myoblasts, a link has been established between mitochondrial morphology and Ca^2+^ uptake. By altering dynamics through infection with adenovirus containing a dominant negative form of DRP1 (DRP1-DN) or silencing MFN2 (MFN2-KD), Kowaltowski et al. induced changes in Ca^2+^ uptake rates and capacity [94]. Notably, the elongated mitochondria showed a greater efficiency and capacity for Ca^2+^ uptake relative to the control. In comparison, MFN2-KD myoblasts had a lower rate and capacity for Ca^2+^ uptake [94].

### 4.2. Mitochondria–Organellar Contact Sites

The signaling mechanisms that are utilized by mitochondrial shaping proteins to mediate crucial cell fate decisions remain to be uncovered. It is known that intraorganellar contacts between the mitochondrial network, ER, and nucleus exist. Instances where mitochondria localize adjacent to the ER and nucleus and interact with them could be areas where potential crosstalk or metabolite exchange occur. In the context of MuSCs, the role and influence of mitochondrial contact sites with other organelles remains to be uncovered. Interestingly, mitochondria–ER crosstalk is involved with survival during ER stress [95,96]. In a mechanism conserved in many cell types, ER stress is attenuated by the adaptive unfolded protein response (UPR) through molecular pathways such as mTOR, which is involved in mitochondria–ER crosstalk and is necessary for cell survival [97,98]. Similarly, under conditions of mitochondrial dysfunction, the integrated stress response (ISR) serves a key protective role in neural stem cells (NSCs) during neurogenesis [99]. In OPA1-null NSCs, the expression of the transcription factor ATF4, which mediates IRS, was upregulated and was required to rescue NSC proliferation [99]. Additionally, mitochondrial stress also led to the activation of PERK, which is also involved in IRS during ER stress [99,100,101], further linking the mitochondrial state to the ER. Importantly, it is already appreciated that mitochondrial–ER contact is intimately linked with mitochondrial fission [102,103,104]. During DRP1-dependent fission, the ER membrane, through actin dynamics, wraps around the mitochondria, allowing for the formation of a constriction point [102,103,104,105]. Additionally, mitochondria–organellar contact with lipid droplets, peroxisomes, melanosomes, and lysosomes are also reported [106,107,108,109]. One can reasonably postulate that such mechanisms mentioned above may also play crucial roles in MuSC survival and myogenesis, further cementing the importance of mitochondria in MuSC health and function.

### 4.3. Pathological Implications of Imbalanced Mitochondrial Dynamics during Aging

An emerging area of research explores the mechanisms by which mitochondrial dynamics influence tissue health in aging. Interestingly, aged skeletal muscle exhibits a decline of MFN1, MFN2, DRP1, and OPA1 at the transcript and protein levels [110]. MuSCs have been studied in the context of improving the maintenance of aged skeletal muscle for their role in regeneration. In particular, because mitochondrial dynamics drive MuSC activity and fate, rising interest has been found in its role in maintaining muscle health during aging [111]. Chronic MuSC-specific loss of OPA1 leads to hallmarks of stem cell aging, presenting an accelerated aging phenotype [25]. At the forefront of the dysfunction caused by prolonged mitochondrial fragmentation are defects in ROS/GSH signaling and a reduction in both the basal and ATP-linked OCR [25]. OPA1-KO MuSCs are placed in a G_alert_ state of precocious activation; however, they experience proliferation defects, as they fail to meaningfully express the proliferative marker, Ki67, in culture and fail to progress through the R point of the cell cycle. Clusters of chronic OPA1-KO MuSCs are limited to a size of 1 cell after 72 h of culture [25]. Interestingly, the excessive mitochondrial fragmentation observed in aging and in pathologies associated with stem cell exhaustion or depletion have been tied to OPA1 deficiencies [112,113]. A major question that remains to be fully answered is the mechanism by which loss of mitochondrial dynamics regulators occurs during aging, though epigenetic alterations or chromatin remodeling have been proposed [114,115].

Chronic imbalances in mitochondrial dynamics are associated with precocious aging phenotypes in MuSCs and limit their availability to maintain a functional stem cell pool. While it is established that the loss of balanced mitochondrial dynamics impairs the expansion of the MuSC pool and causes cell cycle defects [116], it is not known whether imbalanced mitochondrial dynamics in MuSCs can also cause senescence, a form of irreversible cell cycle arrest and a hallmark of aging. However, it has been shown that other forms of mitochondrial dysfunction in MuSCs, such as defects in the mitochondrial proteome, occur during aging and are associated with a senescent phenotype [117]. Alleviating the defects caused by other forms of mitochondrial dysfunction has been shown to prevent cellular senescence, improve MuSC function with age, and improve the metabolic capacity of geriatric MuSCs [117]. Thus, given that the modulators of mitochondrial dynamics decline in expression levels during aging, it would be of considerable interest to understand their relative contribution to the aged MuSC phenotype and discover whether this can be leveraged to improve the functional capacity of MuSCs and whole skeletal muscle with age.

## 5. Conclusions

Mitochondria are essential metabolic and cell signaling hubs with extensive influence on MuSC activity and progression through the myogenic program. There is a role for transient and temporally regulated imbalances in mitochondrial dynamics in influencing MuSC cell signaling and nuclear gene expression. These coincide with physiological changes in metabolite and redox signaling molecule abundance at each stage of myogenesis, from quiescence to terminal differentiation. By modifying the morphology of the mitochondrial network, changes in metabolism and cell fate occur. The tight regulation of mitochondrial dynamics is essential to MuSC health and function, as a persistent loss of balanced dynamics is a key example of mitochondrial dysfunction and has been extensively associated with pathologies like sarcopenia. The direct manipulation of mitochondrial dynamics must be finely tuned to be used as a direct therapeutic approach so as not to induce further dysregulation. The diverse roles of mitochondrial dynamics, such as interorganellar communication and metabolite exchange, warrant future investigation in order to appreciate their function in MuSCs and to leverage this knowledge for therapeutic approaches.

## Figures and Tables

**Figure 1 cells-13-01773-f001:**
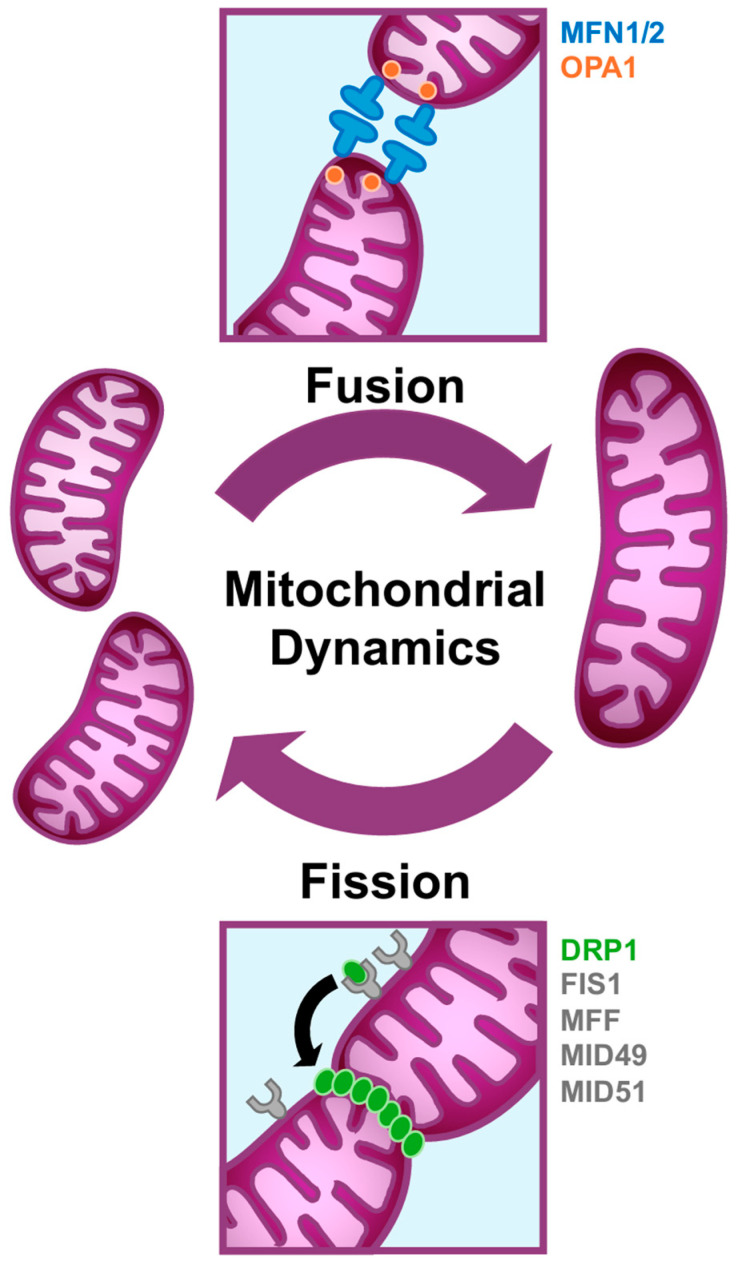
Mitochondrial dynamics are modulated by a family of dynamin-related GTPases. Mitochondria undergo constant cycles of fusion and fission to maintain mitochondrial integrity and to adapt to bioenergetic, metabolic, and cell signaling needs. Fusion of the outer mitochondrial membrane (OMM) is directed by MFN1/2, and the inner mitochondrial membrane (IMM) is mediated by OPA1. Fission occurs at both mitochondrial membranes, directed by active DRP1 and adaptors FIS1, MFF, MID49, and MID51.

**Figure 2 cells-13-01773-f002:**
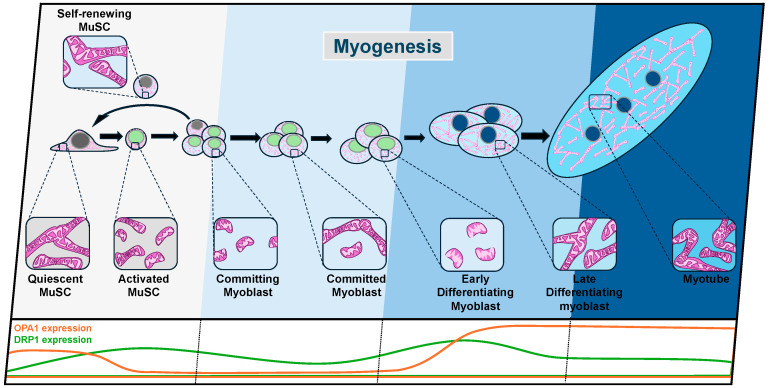
Overview of the morphological changes in mitochondria at each stage of myogenesis. Throughout the myogenic process, MuSCs and myogenic progenitors undergo a significant reorganization of mitochondrial morphology to facilitate the bioenergetic, metabolic, and signaling demands necessary for activation, fate decisions, and differentiation. The differential expression of mitochondrial dynamics regulators, OPA1 and DRP1, is also depicted.

**Figure 3 cells-13-01773-f003:**
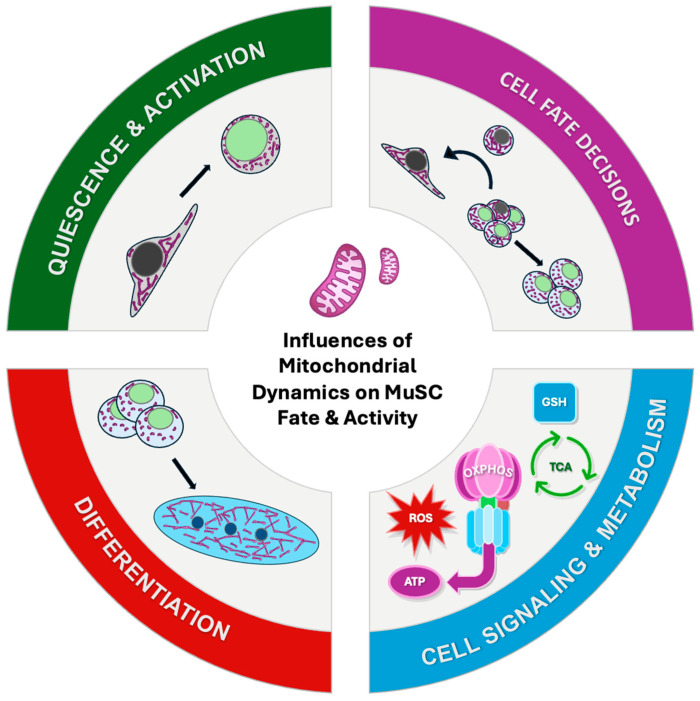
Mitochondrial dynamics have diverse influences on MuSC quiescence, fate, and activity. Mitochondrial dynamics have diverse roles within MuSCs, from regulating the depth of quiescence, promoting activation, regulating fate decisions, and directing commitment to differentiation. At each stage of myogenesis, mitochondrial dynamics influence bioenergetics, metabolism, and cell signaling, among other key processes.

**Table 1 cells-13-01773-t001:** Manipulation of mitochondrial dynamics alters the myogenic program.

Mitochondrial Dynamics Modulator	Manipulation Method	Model System	Influence on the Myogenic Program	References
**OPA1**	OPA1-KO	Primary murine MuSCs	Enhances MuSC activation Enhances MuSC cell cycle entry Increases myogenic commitment Decreases MuSCs self-renewal	[25,58]
	OPA1 siRNA	C2C12 and primary myoblasts	Impairs myogenic differentiation and myotube formation	
	MYLS22			
	OPA1-OE	Primary murine MuSCs	Delays MuSC activation Enhances MuSC self-renewal	
**DRP1**	DRP1-KO	Primary murine MuSCs	Primes MuSC activation Promotes proliferation required for muscle regeneration	[25,59,79,80]
	Low-dose Mdivi-1	Primary murine MuSCs	Reduces myogenic commitment Enhances MuSC self-renewal	
		C2C12 myoblasts	Impairs myogenic differentiation	
**MFN1/2**	Unknown influence in MuSCs and the myogenic program to date			

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
