# Peer review of "Mitochondrial Dynamics Drive Muscle Stem Cell Progression from Quiescence to Myogenic Differentiation"

_cells, 2024, doi:10.3390/cells13211773_

Round 1

Reviewer 1 Report

Comments and Suggestions for Authors

The review by Sommers et al. provides a comprehensive and insightful summary of the current knowledge regarding muscle stem cells (MuSCs) and the critical role that mitochondrial dynamics play in influencing their activity and fate decisions. The authors effectively outline the molecular markers associated with MuSC commitment and follow this with a detailed description of how the mitochondrial network undergoes structural changes during these processes, emphasizing the key proteins involved in mitochondrial dynamics.

Along the manuscript, the authors shift focus to examine how the genetic and chemical regulation of mitochondrial fusion and fission affects MuSC commitment and differentiation. This section explores the impact of modulating mitochondrial dynamics through key regulators such as OPA1, MFNs, and DRP1. The authors also delve into the molecular pathways associated with glutathione (GSH), nitric oxide (NO), and other signaling molecules, linking these pathways to both differentiation and mitochondrial dynamics.

While the manuscript is well-organized and the content is presented in a logical manner, I find the figures included in the review to be less helpful in enhancing the reader’s understanding of the complex regulatory mechanisms discussed. I recommend the authors consider developing a new figure or table that summarizes the specific effects of manipulating key proteins involved in mitochondrial dynamics (e.g., OPA1 knockdown (KD), overexpression (OE), MFN1 knockout (KO), DRP1 KO, DRP1 OE, etc.) on cell fate decisions and metabolic changes. This figure could highlight known outcomes for each manipulation, and mark areas where information is currently lacking with an "unknown" label.

Additionally, Figure 2 could be improved by visually representing the changes in fusion and fission protein levels throughout different stages of myogenesis. It would also be valuable to indicate whether the knockdown or overexpression of these proteins influences differentiation at specific stages, either blocking or promoting progression. This would provide a clearer picture of how mitochondrial dynamics regulate myogenic differentiation.

I hope the authors will find the suggestions to improve the manuscript helpful.

Author Response

Response to Reviewer 1 Comments

Reviewer #1:

Comment #1: While the manuscript is well-organized and the content is presented in a logical manner, I find the figures included in the review to be less helpful in enhancing the reader’s understanding of the complex regulatory mechanisms discussed. I recommend the authors consider developing a new figure or table that summarizes the specific effects of manipulating key proteins involved in mitochondrial dynamics (e.g., OPA1 knockdown (KD), overexpression (OE), MFN1 knockout (KO), DRP1 KO, DRP1 OE, etc.) on cell fate decisions and metabolic changes. This figure could highlight known outcomes for each manipulation, and mark areas where information is currently lacking with an "unknown" label.

Response: We thank the reviewer for this suggestion. We have now added a table (new Table 1) that summarizes the specific effects of manipulating key proteins involved in mitochondrial dynamics, including OPA1, DRP1 and MFN1/2 on muscle stem cell fate decisions.

Comment #2: Additionally, Figure 2 could be improved by visually representing the changes in fusion and fission protein levels throughout different stages of myogenesis. It would also be valuable to indicate whether the knockdown or overexpression of these proteins influences differentiation at specific stages, either blocking or promoting progression. This would provide a clearer picture of how mitochondrial dynamics regulate myogenic differentiation.

Response: As per the Reviewer’s suggestion, we have amended Figure 2 (new Figure 2) to include changes in OPA1 and DRP1 protein levels throughout different stages of myogenesis. In addition, we have included how the knockdown or overexpression of these proteins influences differentiation at specific stages in the new Table 1.

Reviewer 2 Report

Comments and Suggestions for Authors

Review MuSC

The review focuses on the role of mitochondrial dynamics in muscle stem cell differentiation, which provides an interesting and relevant specific perspective. The authors provide valuable information. However, the title promises a broader content that includes mitochondrial signalling and metabolism, and should perhaps focus on the ‘Role of mitochondrial dynamics in muscle stem cell differentiation’.

Else, I have only minor comments and recommend publication after clarifying the issues.

P1, L33-L39: MyoG is introduced with two different names, clarify the nomenclature of the genes involved in myoblast/myotube differentiation, in line 33, MyoD is the myogenic differentiation factor, not MyoG

P5, L157: Mdivi-1 is mean while recognized as a complex I inhibitor (1, 2)

P5, L179-177: check redundancy with previous paragraphs

1.           E. A. Bordt et al., The Putative Drp1 Inhibitor mdivi-1 Is a Reversible Mitochondrial Complex I Inhibitor that Modulates Reactive Oxygen Species. Dev Cell 40, 583-594 e586 (2017).

Author Response

Response to Reviewer 2 Comments

Reviewer #2:

Comment #1: The review focuses on the role of mitochondrial dynamics in muscle stem cell differentiation, which provides an interesting and relevant specific perspective. The authors provide valuable information. However, the title promises a broader content that includes mitochondrial signalling and metabolism, and should perhaps focus on the ‘Role of mitochondrial dynamics in muscle stem cell differentiation’.

Response: We thank the reviewer for this thoughtful comment and agree that the title was over ambitious. We have now changed the title to “Mitochondrial dynamics drive muscle stem cell progression from quiescence to myogenic differentiation”.

Comment #2: P1, L33-L39: MyoG is introduced with two different names, clarify the nomenclature of the genes involved in myoblast/myotube differentiation, in line 33, MyoD is the myogenic differentiation factor, not MyoG.

Response: We apologize for the confusion. We have now modified the text and removed “differentiation mediator” to avoid any misunderstanding.

Comment #3: P5, L157: Mdivi-1 is mean while recognized as a complex I inhibitor (1, 2)

Response: We would like to clarify that Mdivi-1 is recognized as a Complex 1 inhibitor when it is used in high doses (20-50mM). In the study by Baker et al. the dose of Mdivi-1 used (2mM) is at the optimal concentration to effectively inhibit DRP1, without affecting Complex 1. We had added the following sentence to clarify this point:

P5, L167-169; “Of note, it is important to consider that Mdivi-1 at high doses is a complex-1 inhibitor 63. Therefore, only low doses can be utilized for proper evaluation of the effects of mitochondrial fission inhibition in which Baker et al. respected 25, 63

Comment #4: P5, L179-177: check redundancy with previous paragraphs

Response: We have read the lines outlined by the Review but have not noticed any obvious redundancy with other paragraphs.